# An unconventional proanthocyanidin pathway in maize

Nan Lu[1], Ji Hyung Jun[1,2], Ying Li[1] & Richard A. Dixon [1] ✉

Proanthocyanidins (PAs), flavonoid polymers involved in plant defense, are also beneficial to human health and ruminant nutrition. To date, there is little evidence for accumulation of PAs in maize (*Zea mays*), although maize makes anthocyanins and possesses the key enzyme of the PA pathway, anthocyanidin reductase (ANR). Here, we explore whether there is a functional PA biosynthesis pathway in maize using a combination of analytical chemistry and genetic approaches. The endogenous PA biosynthetic machinery in maize preferentially produces the unusual PA precursor (+)-epicatechin, as well as 4*β*-(S-cysteinyl)-catechin, as potential PA starter and extension units. Uncommon procyanidin dimers with (+)-epicatechin as starter unit are also found. Expression of soybean (*Glycine max*) anthocyanidin reductase 1 (ANR1) in maize seeds increases the levels of 4*β*-(S-cysteinyl)-epicatechin and procyanidin dimers mainly using (-)-epicatechin as starter units. Introducing a *Sorghum bicolor* transcription factor (SbTT2) specifically regulating PA biosynthesis into a maize inbred deficient in anthocyanin biosynthesis activates both anthocyanin and PA biosynthesis pathways, suggesting conservation of the PA regulatory machinery across species. Our data support the divergence of PA biosynthesis across plant species and offer perspectives for future agricultrural applications in maize.

Proanthocyanidins (PAs), or condensed tannins, are the second most abundant plant phenolic compounds after lignin[1]. They are oligomers and polymers of flavan-3-ols and are produced in several tissues of vascular plants, providing protection from herbivores, fungal pathogens, and ultraviolet radiation[2]. The antibacterial activities of PAs and their precursor flavan-3-ols and their beneficial effects in preventing cardiovascular disease make these compounds popular health supplements and targets for increasing food nutritional value[3–5]. Addition of PAs to ruminant animal feed can improve nitrogen retention in the rumen and reduce pasture bloat caused by production of methane gas, improving animal health and productivity and reducing greenhouse gas emissions from livestock[6,7]. Compared to the well-characterized flavan-3-ol and PA biosynthesis pathways in model species such as *Arabidopsis thaliana* and *Medicago truncatula*, the biosynthetic routes to PAs and their building blocks in monocotyledonous crops remain largely unexplored[8].

As the most important cereal crop cultivated worldwide, maize (*Zea mays*) provides food, animal feed, and biofuels[9,10]. Through domestication, breeders have generated a large collection of maize cultivars with distinct seed pigmentation, determined by the accumulation of different types and levels of flavonoids, including anthocyanins, quercetin, maysin, and phlobaphenes[11]. In maize seeds, anthocyanins mainly accumulate in aleurone tissues[12], where their biosynthesis has been extensively studied and the enzymes involved identified and characterized[13]. The transcriptional regulation of anthocyanin biosynthesis depends on a ternary complex of transcription factors, namely MYB-bHLH-WD40. Two anthocyanin-related MYB transcription factors, *C1* and *Pl*, are involved in tissue-specific anthocyanin deposition in maize[14–17]. Besides anthocyanins, phlobaphenes are detected in pericarp tissues of some maize varieties with brick-red seeds. The biosynthesis of phlobaphenes is regulated by an

[1]BioDiscovery Institute and Department of Biological Sciences, University of North Texas, Denton, TX 76203, USA. [2]Present address: Children's Research Institute and the Department of Pediatrics, University of Texas Southwestern Medical Center, Dallas, TX 75390, USA. ✉e-mail: Richard.Dixon@unt.edu

R2R3-type MYB transcription factor, *pericarp color 1* (*P1*)[18]. The key enzyme for phlobaphene biosynthesis, a bifunctional dihydroflavonol 4-reductase/flavanone 4-reductase, encoded by the *A1* locus, is induced by *P1* and diverts the flux of substrates from anthocyanin biosynthesis toward the phlobaphene pathway. Homologs of *P1* and *A1* have been identified in *Sorghum bicolor*, suggesting a conserved regulatory mechanism for the phlobaphene pathway[19]. The biosynthesis of anthocyanins, phlobaphenes and PAs shares common precursors and intermediates (Supplementary Fig. 1a). However, few reports exist on the investigation of PAs and their precursors in maize, and, where available, suggest only very low levels[20] or the existence of unusual flavan-3-ol anthocyanin conjugates in some lines[21].

A key enzyme in the PA biosynthesis pathway, anthocyanidin reductase (ANR), converts anthocyanidins to the flavan-3-ol building blocks of PAs (i.e., catechins and epicatechins)[22,23] (Supplementary Fig. 1b, c). Loss-of-function of ANR leads to reduced levels of PAs and increased anthocyanins in *Arabidopsis thaliana* seeds. Maize ANR (ZmANR1), although able to convert cyanidin to catechin and epicatechin in vitro, does so at a much lower rate than the reaction catalyzed by ANRs from the dicots soybean (*Glycine max*), *Medicago truncatula*, or *A. thaliana*, and the major product of ZmANR1 is (+)-epicatechin rather than the typical (-)-epicatechin produced by other ANRs[23] (Supplementary Fig. 1b, c). When ZmANR1 is ectopically expressed in the *A. thaliana ANR* mutant *ban*, the epicatechin level is slightly increased in developing seeds, but procyanidin B2 dimer is not detected, suggesting that ZmANR1 from maize is functionally distinct from the ANRs of *A. thaliana* and other PA-rich dicot plants[23].

Like those in the anthocyanin pathways, many PA-related enzymes are transcriptionally regulated by the ternary MBW complex consisting of MYB (TT2 or MYB5-type), bHLH (TT8) and WD40 transcription factors[24]. The maize bHLH family transcription factors *Lc* and *Sn* were able to induce the production of anthocyanins and PAs when ectopically expressed in alfalfa (*Medicago sativa*) and lotus (*Nelumbo nucifera*)[25,26]. In addition, ectopic expression of *pale aleurone color 1* (*Pac1*), a maize WD40 family transcription factor, in the *A. thaliana ttg11* mutant restored the levels of both anthocyanins and PAs[27]. Therefore, some enzymes and transcription factors for PA biosynthesis may exist in maize, but their in vivo functions remain unclear in light of the general lack of long chain oligomeric PAs in this species.

Here, we probe the occurrence and diversity of PAs and their precursors in different maize varieties and explore the consequences of expressing heterologous ANR and TT2 homologs on production of PAs or their biosynthetic intermediates in these lines. Our results provide *in planta* evidence for a PA pathway in maize that generates unconventional monomers and their conjugates but does not support efficient polymerization to the long chain forms that improve the nutritional quality of feeds. The implications of these findings for our understanding of PA assembly are discussed.

## Results

### Mining PA-related genes in monocotyledonous plants

To interrogate the potential for PA biosynthesis in maize, we searched for homologs of genes known to be involved in PA biosynthesis in other plant species. *Anthocyanidin Reductase* (*ANR*) encodes an enzyme functionally conserved across species that is essential for PA biosynthesis. Using GmANR1 to BLAST against the maize genome, we found two strong candidate *ANR* genes, namely *ZmANR1* (GRMZM2g097854) and *ZmANR2* (GRMZM2g097841), consistent with earlier predictions[18]. These two genes locate in close proximity to each other on chromosome 10. Transcriptional analysis of these two *ANR* homologs in colored maize seeds of varieties Suntava (ST, purple seeds) and Black Mexican (BM, blue/black seeds) indicated that *ZmANR1* is expressed at a much higher level than *ZmANR2* in developing seeds (14 days-after-pollination, DAP) (Fig. 1a). Subcellular localization experiments using Arabidopsis protoplasts showed that

ZmANR1 localizes to the cytosol, similar to soybean GmANR1 (Fig. 1b). Ectopic expression of *ZmANR1* in the *A. thaliana ban* mutant led to increased levels of free epicatechin but no procyanidin B2 dimers [(-)-epicatechin-(-)-epicatechin][23]. To test whether ZmANR differs from dicot ANRs in its ability to direct PA formation in planta, we expressed ZmANR1, *A. thaliana* ANR (AtANR) and GmANR1 in tobacco (*Nicotiana tabacum* cv *Xanthi*) overexpressing the *A. thaliana PAP1* (*PRODUCTION OF ANTHOCYANIN PIGMENT 1*) gene. PAP1 is a conserved MYB transcription factor regulating anthocyanin biosynthesis in plants[28]. These PAP1-OX tobacco plants produce high levels of anthocyanins, resulting in red-colored petals[22], and overexpression of AtANR or GmANR1 in the PAP1-OX background diverted flavonoid precursors from anthocyanins toward PA biosynthesis and consequently loss of the red floral pigmentation (Fig. 1c). However, PAP1-OX/ZmANR1 lines displayed red-colored petals similar to PAP1-OX (Fig. 1c). Staining with *p*-dimethylaminocinnamaldehyde (DMACA) showed that PAs accumulated in petals of PAP1-OX/AtANR and PAP1-OX/GmANR1 plants, as indicated by the purple color, but not in flowers of PAP1-OX/ZmANR1 or PAP1-OX plants. Further analyses of PAs using liquid chromatography-mass spectrometry (LC-MS) showed that both epicatechin and procyanidin B2 dimers [(-)-epicatechin-(-)-epicatechin] accumulated in PAP1-OX/AtANR and PAP1-OX/GmANR1 flowers, whereas only a small increase of epicatechin but not procyanidin B2 was observed in PAP1-OX/ZmANR1 (Fig. 1d). Furthermore, cysteinyl-epicatechin, an extension unit for PA polymerization, was substantially accumulated in PAP1-OX/AtANR and PAP1-OX/GmANR1 lines, but was only slightly elevated in the PAP1-OX/ZmANR1 line with similar level of transgene expression (Fig. 1d, e).

Leucoanthocyanidin reductase (LAR) generates catechin from leucocyanidin and also participates in PA polymerization in some species by controlling the ratio of PA starter (epicatechin) and extension (cysteinyl-epicatechin) units[29] (Supplementary Fig. 1a). Searching the maize and sorghum genomes using MtLAR from *M. truncatula* and VvLAR1 from grapevine (*Vitis vinifera*) returned no apparent LAR homolog, with an annotated 2´-hydroxyisoflavone reductase (Zm00001d040173) and an uncharacterized protein (Sb03g043200) showing the highest similarity in maize and sorghum, respectively. At least one OsLAR (LOC_Os03g15360) was present in the rice genome. Further sequence analysis showed that Sb03g043200 and LOC_Os03g15360 lack the ICCNSIA and THDIFI domains conserved in functional LAR homologs[30] and share no obvious sequence similarity with LAR homologs in other plant species (Supplementary Fig. 2). Thus, sorghum and maize appear not to contain apparent LAR homologs.

TT8 (bHLH) and TTG1 (WD40) transcription factors are required for expression of PA biosynthesis in *A. thaliana* and *M. truncatula*[31,32]. However, no apparent TT2-type MYB has yet been identified in maize[33,34]. The maize MYB transcription factors with the highest sequence similarity to TT2 homologs in *A. thaliana* (AtTT2) and *M. truncatula* (MtMYB14) are *C1* and *Pl*, both of which have been identified as anthocyanin regulators[14,15,34]. Further sequence analysis showed that both rice and sorghum have both C1- and TT2-type MYBs, i.e., OsC1 (LOC_Os06g10350), OsMYB3/OsTT2 (LOC_Os03g29614), SbC1 (Sb10g006700), and SbTT2 (Sb01g032770) (Supplementary Fig. 3a, b). Phylogenetic analysis indicated that the TT2 homologs from sorghum, rice, *M. truncatula* and *A. thaliana* belong to a clade distinct from C1-type MYBs (Supplementary Fig. 3c). MYB5-type transcription factors also play important roles in plant development and PA biosynthesis in dicot species[35], and homologous genes can be identified in the maize, rice, and sorghum genomes (Supplementary Fig. 3d). Taken together, PA-rich monocots including sorghum and rice possess C1-type, TT2-type and MYB5-type MYBs, whereas maize only appears to contain the C1- and MYB5-type.

Overall, these results indicate that maize possesses neither an effective ANR or LAR that facilitate PA biosynthesis nor a TT2-family transcription factor to regulate expression of their genes.

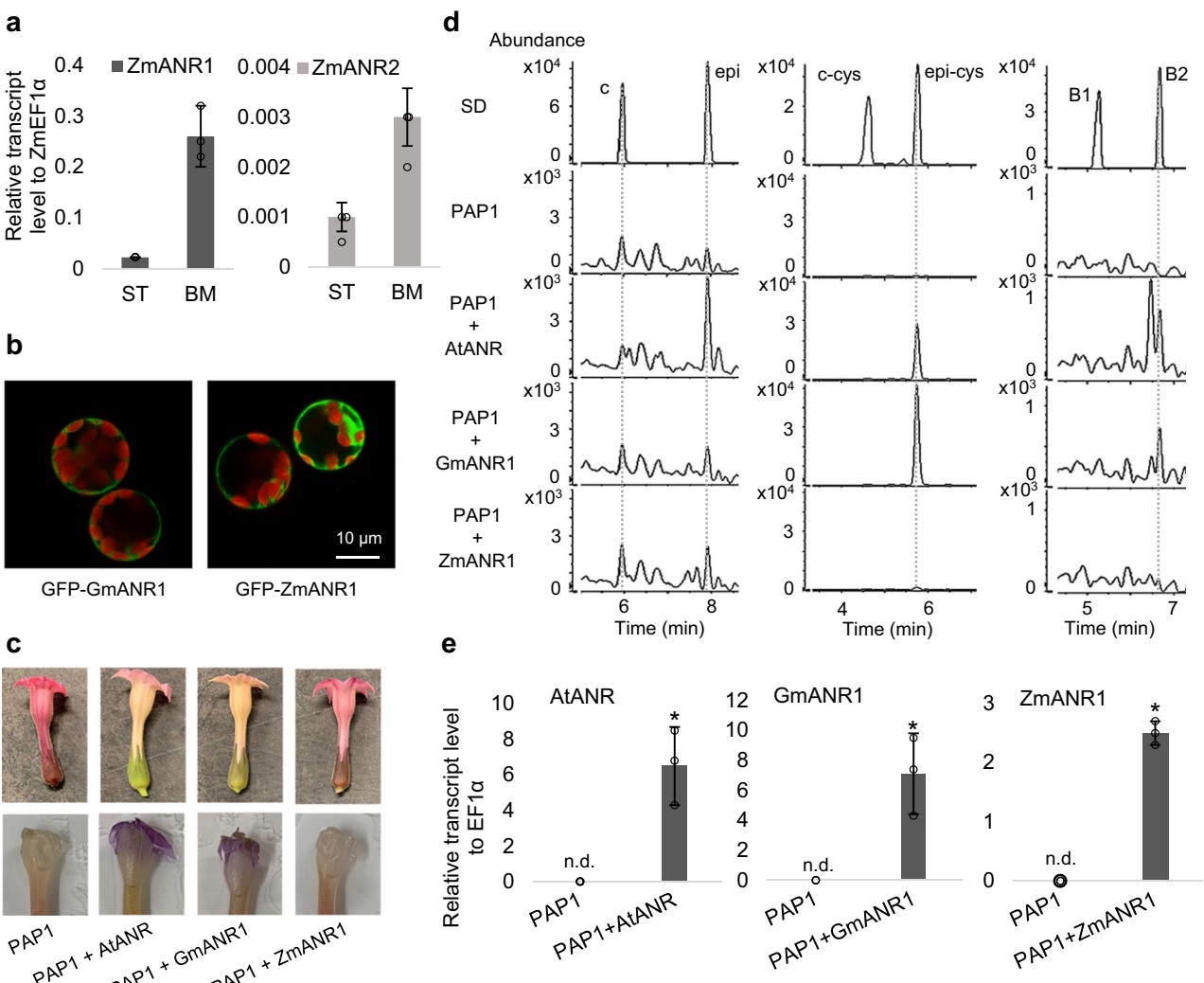

**Fig. 1 | Ectopic expression of ZmANR1 fails to generate PA precursors in high anthocyanin producing tobacco. a** Transcript levels of *ZmANR1* and *ZmANR2* in developing seeds (14-days after pollination) of maize cultivars ST (Suntava) and BM (Black Mexican). Data are presented as mean ± S.D. (*n* = 3, independent biological replicates). Differences in transcript levels of *ZmANR1* and *ZmANR2* between ST and BM maize are significant (*P* < 0.01) as determined by two-tailed Student's *t*-test. *ZmEF1α* was used as the reference gene. **b** Confocal microscopy images of GFP-tagged GmANR1 and ZmANR1 showing their subcellular localization in *A. thaliana* protoplasts. GFP-tagged GmANR1 and ZmANR1 appear green. Autofluorescence of chloroplasts appears red. Images are representative of three independent replicates. **c** Images of tobacco flowers expressing *PAP1*, *PAP1* with *AtANR*, *PAP1* with *GmANR1*, and *PAP1* with *ZmANR1* before (top) and after (bottom) DMACA staining. The distribution of PAs in tobacco flowers is indicated as purple coloration after DMACA staining. *AtANR* and *GmANR* were cloned from *Arabidopsis thaliana* (Col-0) and *Glycine max* (cv *Clark*), as described[23] (**d**) Selected ion chromatogram of catechin and epicatechin (left, m/z = 289.0718 ± 10 ppm), 4β-(S-cysteinyl)-catechin and 4β-(S-cysteinyl)-epicatechin (middle, m/z = 408.0759 ± 10 ppm), procyanidin dimers B1 and B2 (right, m/z = 577.1360 ± 10 ppm) in PA extracts from tobacco flowers. **e** Transcript levels of *AtANR*, *GmANR1* and *ZmANR1* in tobacco plants expressing *PAP1*, *PAP1* with *AtANR*, *PAP1* with *GmANR1*, and *PAP1* with *ZmANR1*. Data are presented as mean ± S.D. (*n* = 3, independent biological replicates). Asterisks indicate significant difference relative to the untransformed control at *P* < 0.01 as determined by Student's *t*-test. Transcripts not detected are labeled as n.d. *NtEF1α* was used as the reference gene. Source data for Fig. 1a and Fig. 1e are provided in the Source Data file.

## Occurrence and diversity of PA precursors in different maize varieties

Given the lack of information regarding the presence or levels of flavan-3-ols and other PA-related metabolites in maize, we first searched for these metabolites in seeds of various maize varieties. In yellow maize seeds (FBLL), only trace amounts of epicatechin were detected (Fig. 2, Supplementary Fig. 4), whereas in brown (Osage, OSA), black (Black Mexican, BM), and purple (Suntava, ST; Arequipa, AREQ) seeds, larger amounts of catechin and epicatechin were detected, with more epicatechin than catechin (Fig. 2), consistent with the demonstration that ZmANR1 preferentially produces epicatechin over catechin[23]. The PA extension units cysteinyl-(epi)catechin were detected in brown (Osage, OSA), black (Black Mexican, BM), and purple (Suntava, ST) seeds, with the level of 4β-(S-cysteinyl)-catechin generally higher than

that of 4β-(S-cysteinyl)-epicatechin, or, in the case of AREQ, exclusively 4β-(S-cysteinyl)-catechin being present (Fig. 2). The levels of the anthocyanin cyanidin 3-*O*-glucoside were generally negatively correlated with free epicatechin levels in OSA, BM, ST and AREQ seeds (Fig. 2). In maize seeds naturally accumulating phlobaphenes, no (epi) catechin or cysteinyl-(epi)catechin were detected (Supplementary Fig. 5). In sorghum seeds included as a positive control, catechin and procyanidin B3 (catechin dimer) were observed (Supplementary Fig. 6).

To compare the PA profile of maize with that from a typical PA-rich crop, we analyzed the soluble PA fraction extracted from seeds of soybean, in which epicatechin was the primary flavan-3-ol monomer, and B2 [(-)-epicatechin-(-)-epicatechin] was the main procyanidin dimer (Supplementary Fig. 7). In contrast to maize, the PA extension

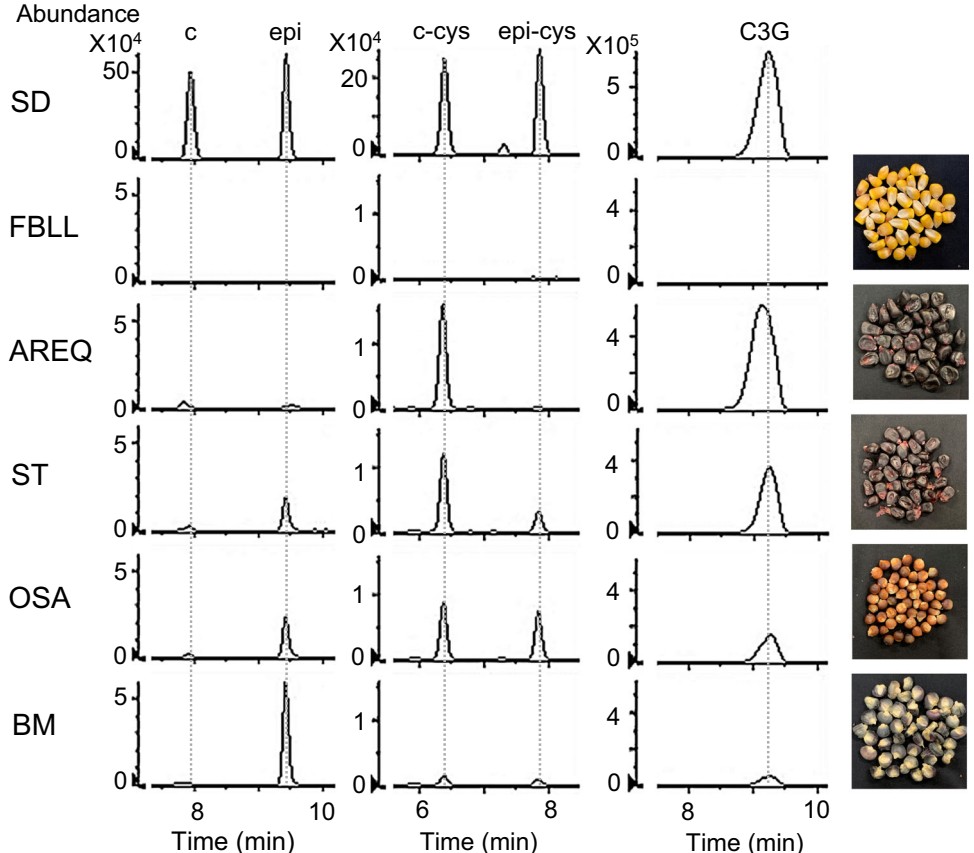

**Fig. 2 | Analysis of PA starter and extension units and anthocyanins in maize seeds.** Left panels, selected ion chromatograms of catechin and epicatechin (m/z = 289.0718 ± 10 ppm). Middle panels, selected ion chromatograms of 4$\beta$-(S-cysteinyl)-catechin and 4$\beta$-(S-cysteinyl)-epicatechin (m/z = 408.0759 ± 10 ppm). Right panels, selected ion chromatograms of cyanidin 3-O-glucoside (m/z = 447.0940 ± 10 ppm). SD, chemical standards; c, catechin; epi, epicatechin; c-cys, cysteinyl-catechin; epi-cys, cysteinyl-epicatechin; C3G, cyanidin 3-O-glucoside. Maize varieties are FBLL; AREQ, Arequipa; ST, Suntava; OSA, Osage; BM, Black Mexican.

unit 4$\beta$-(S-cysteinyl)-epicatechin was detected in developing seeds but not in mature seeds (Supplementary Fig. 7).

**Expression of *GmANR1* leads to accumulation of 4$\beta$-(S-cysteinyl)-epicatechin in maize seeds with active flavonoid biosynthesis**

ANR participates in the generation of both starter and extension units in PA-rich plants[23]. To test whether the ANR from a PA-rich species can boost PA production in maize, we introduced soybean GmANR1 into HiII maize that only produces trace amounts of epicatechin and no cysteinyl-epicatechin extension units. We selected HiII for transformation because of its high transformation efficiency and short time needed for plant regeneration. Three independent GmANR1-OX lines with confirmed *GmANR1* expression were selected for further biochemical analyses (Supplementary Fig. 8a, b). The contents of anthocyanins and soluble PAs were low and not significantly different between seeds of untransformed plants and the three GmANR1-OX lines. Epicatechin levels were similar between untransformed and GmANR1-OX seeds, and no 4$\beta$-(S-cysteinyl)-catechin or 4$\beta$-(S-cysteinyl)-epicatechin were detected (Supplementary Fig. 8c). We reasoned that the PA biosynthesis pathway may be substrate-limited since the general flavonoid pathway is inactive in white- or yellow-colored seeds such as HiII. To provide more substrate for PA biosynthesis, we crossed GmANR1-OX (line GmANR1-24) with Suntava (ST), an anthocyanin-rich purple maize, and backcrossed the F$_1$ plants for two more generations using ST as the recurrent parent to obtain BC$_2$ seeds. The content of 4$\beta$-(S-cysteinyl)-epicatechin in ST-GmANR1 BC$_2$ transgenic seeds was increased by up to 19-fold compared with that in untransformed ST seeds. In addition, the ratio of 4$\beta$-(S-cysteinyl)-epicatechin to 4$\beta$-(S-cysteinyl)-catechin was increased from 0.3 in untransformed seeds to 8 in ST-GmANR1 seeds (Fig. 3a). These data are consistent with previous studies indicating that GmANR1 is involved in the formation of 4$\beta$-(S-cysteinyl)-epicatechin and that ZmANR1 may not be able to efficiently catalyze the conversion of flaven-3,4-diol to 2,3-*cis*-leucocyandin and 4$\beta$-(S-cysteinyl)-epicatechin (Fig. S1)[23]. The lower expression level of endogenous *ZmANR1* compared to ectopically expressed *GmANR1* may partially contribute to different PA levels in different maize lines. Further phloroglucinolysis of PA polymers showed that ST-GmANR1 had higher levels of released (epi)catechin-phloroglucinol (representing PA extension units from polymers) than either parental line (Fig. 3b).

To confirm that the increased 4$\beta$-(S-cysteinyl)-epicatechin in ST-GmANR1 is independent of the genetic background that introduces an active flavonoid pathway, we crossed GmANR1-24 (HiII) with another anthocyanin-rich maize, Osage (OSA), and backcrossed the F$_1$ generation with OSA as recurrent parent to obtain BC$_1$ seeds. Similar to the effect of GmANR1-OX in the ST background, the level of 4$\beta$-(S-cysteinyl)-epicatechin and the ratio of 4$\beta$-(S-cysteinyl)-epicatechin to 4$\beta$-(S-cysteinyl)-catechin were increased by approximately 6-fold and 9-fold, respectively, in OSA-GmANR1 BC$_1$ seeds compared with untransformed OSA seeds (Fig. 3c), indicating that the altered composition of soluble PA precursors was caused by *GmANR1* expression. Our results highlight the endogenous ANR as a bottleneck for PA biosynthesis in maize due to its inability to facilitate formation of extension units.

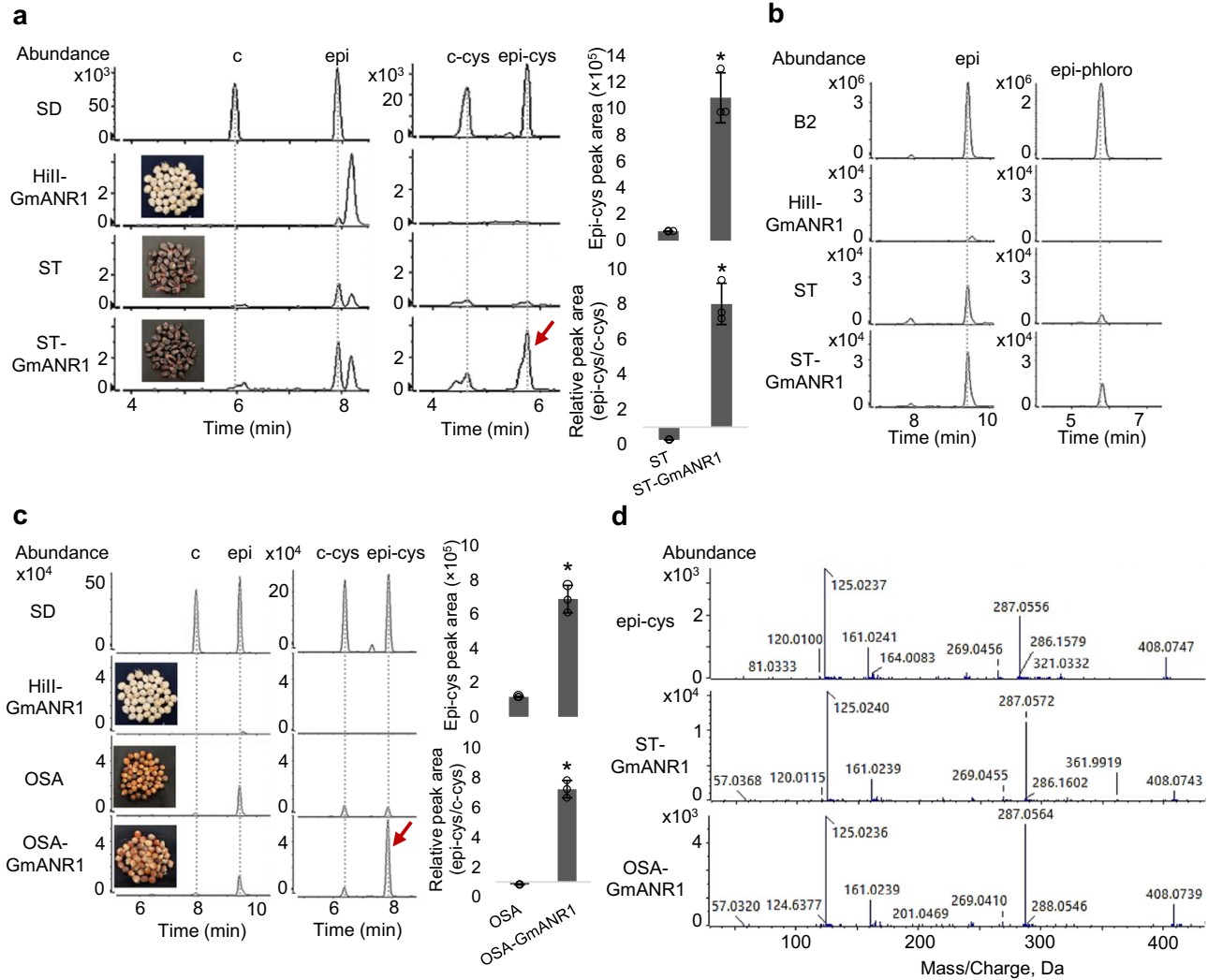

**Fig. 3 | Formation of PA precursors in different maize varieties expressing GmANR1.** Selected ion chromatograms of catechin and epicatechin (m/z = 289.0718 ± 10 ppm), as well as 4β-(S-cysteinyl)-catechin and 4β-(S-cysteinyl)-epicatechin (m/z = 408.0759 ± 10 ppm) in seeds of untransformed and transgenic ST (**a**) and OSA (**c**) maize. Peak areas of 4β-(S-cysteinyl)-epicatechin and ratio of 4β-(S-cysteinyl)-epicatechin/4β-(S-cysteinyl)-catechin in ST, ST-GmANR1, OSA and OSA-GmANR1 are shown in histograms. Data are presented as mean ± S.D. (*n* = 3, independent biological replicates). Asterisks denote significant difference relative to wild-type ST or OSA at *P* < 0.01 determined by two-tailed Student's *t*-test. **b** Phloroglucinolysis of PAs extracted from seeds of ST, ST-GmANR1 and HilI-

GmANR1 using procyanidin dimer B2 as standard. Left, selected ion chromatograms of epicatechin monomers (m/z = 289.0718 ± 10 ppm); right, selected ion chromatograms of epicatechin-phloroglucinol (m/z = 413.0876 ± 10 ppm). **d** MS/MS spectra of 4β-(S-cysteinyl)-epicatechin in the standard and PAs extracted from seeds expressing *GmANR1*. SD, chemical standards; ST, Suntava maize; OSA, Osage maize; HilI-GmANR1, HilI maize expressing *GmANR1* (line GmANR1-24); ST-GmANR1, seeds from genetic cross between ST and GmANR1-24; OSA-GmANR1, seeds from genetic cross between OSA and GmANR1-24. Red arrows indicate peak representing cysteinyl-epicatechin. Source data for Fig. 3a and Fig. 3c are provided in the Source Data file.

## Expressing *SbTT2* or *SbMYB5* in yellow-seed maize activates both the anthocyanin and PA precursor pathways

Since maize does not have an apparent homolog of TT2, we reasoned that, in addition to the issue of ZmANR specificity, maize may lack a functional transcription factor to activate the PA biosynthesis pathway. Sorghum is a PA-rich monocot and a close-relative to maize, so we investigated whether SbTT2 and SbMYB5 could be used to activate PA accumulation in maize. Transactivation assays using Arabidopsis protoplasts indicated that both SbTT2 and SbMYB5, when combined with GmTT8 and GmWD40, were able to activate the *GmANR1* promoter thereby driving firefly *luciferase* gene expression (Supplementary Fig. 9a), suggesting that SbTT2 and SbMYB5 may function in a manner similar to that of GmTT2 and GmMYB5 in soybean and related MYBs in other species[35–37].

To first test the in vivo activity of the transcription factors, *SbTT2*, *SbMYB5* and *GUS* (control) were individually expressed in

soybean hairy roots. DMACA-stained PAs (purple color) were observed in vascular tissues of SbTT2 roots but were undetectable in SbMYB5 or GUS expressing roots (Supplementary Fig. 9b). Quantitative reverse transcription (qRT)-PCR showed that transcript levels of *GmANR1*, *GmLAR2* and *GmANS* were significantly higher in SbTT2 and SbMYB5 lines than in GUS lines, with *GmANR1* primarily induced by SbTT2 and *GmLAR2* more effectively induced by SbMYB5 (Supplementary Fig. 9c). Phloroglucinolysis analysis showed increased levels of oligomer/polymer-derived epicatechin-phloroglucinol in SbTT2-expressing compared with the GUS control (Supplementary Fig. 9d). These results suggest that, while both SbTT2 and SbMYB5 can induce the expression of PA-related genes, SbTT2 is the more effective in activating the PA biosynthesis/polymerization pathway, at least in soybean.

SbTT2 and SbMYB5 were then introduced individually into B104 maize, an inbred deficient in *C1* and *B-peru* activities, transcription

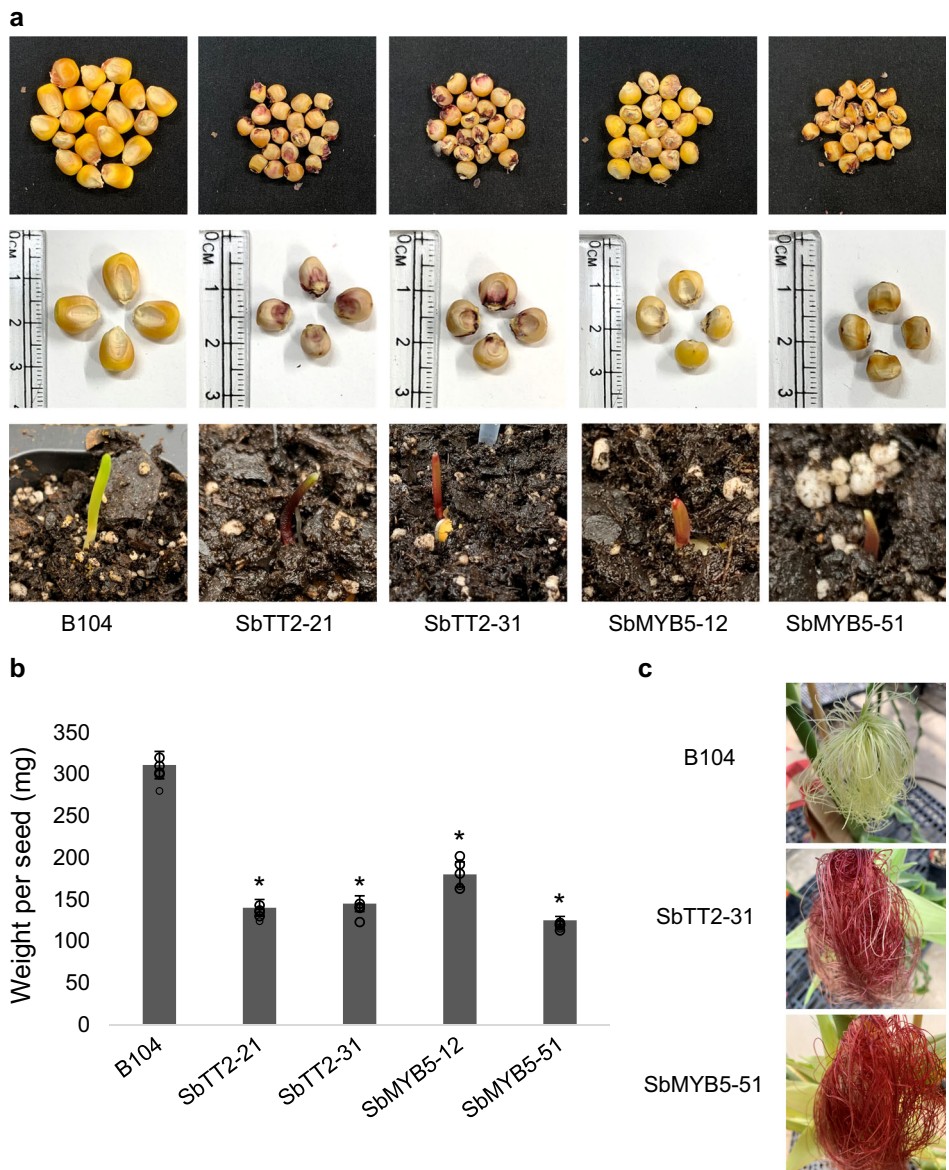

**Fig. 4 | Expression of SbTT2 or SbMYB5 reduces growth and enhances anthocyanin accumulation in yellow-seeded maize. a** Top, images of seeds from untransformed B104, transgenic lines expressing *SbTT2* (SbTT2-21 and SbTT2-31), and transgenic lines expressing *SbMYB5* (SbMYB5-12 and SbMYB5-51); middle, zoom-in images of seeds in the top row showing differences in seed size and color; bottom, images of maize coleoptile at 5 days after seed germination showing differences in color. **b** Seed weight of untransformed B104 and transgenic lines expressing *SbTT2* or *SbMYB5*. Data are presented as mean ± S.D. (*n* = 6, independent biological replicates). Asterisks denote significant difference relative to untransformed B104 at *P* < 0.01 determined by two-tailed Student's *t*-test. **c** Images of maize silk showing differences in color. Source data for Fig. 4b are provided in the Source Data file.

factors regulating anthocyanin biosynthesis[38]. Two independent lines of SbTT2-OX (SbTT2-21 and SbTT2-31) and two independent lines of SbMYB5-OX (SbMYB5-51 and SbMYB5-12) were selected for further analyses (Fig. 4a). Seeds from both SbTT2-OX and SbMYB5-OX lines were smaller than untransformed seeds (Fig. 4a) and weighed less (Fig. 4b). In contrast to the yellow seed color of the untransformed maize (B104), seeds of the two SbTT2-OX lines were purple, and seeds from SbMYB5-OX lines were slightly darker than untransformed seeds (Fig. 4a). Anthocyanin-related pigmentation was also observed in the coleoptile tissue and later in silks (Fig. 4a [lower panels] and 4c). Qualitative analysis of anthocyanins using high performance LC (HPLC) and LC-MS indicated that the anthocyanins produced in SbTT2-OX and SbMYB5-OX are predominantly cyanidin-based and include cyanidin 3-*O*-glucoside (Supplementary Fig. 10 a–c). Quantification of total anthocyanins showed that expression of *SbTT2* or *SbMYB5* induced anthocyanin biosynthesis in maize seeds, and the level of

anthocyanin was higher in SbTT2-OX than in SbMYB5-OX (Supplementary Fig. 10d).

Transcript analysis showed that expression of *ZmANR1* and *ZmANR2* was up-regulated in developing seeds of SbTT2-OX and SbMYB5-OX, with *ZmANR1* transcripts to a much higher level than *ZmANR2* (Fig. 5a). Both soluble and insoluble PA levels were increased, by approximately 10-15-fold and 4-8-fold, in seeds of SbTT2-OX and SbMYB5-OX, respectively, compared to the untransformed maize (Fig. 5b), although the absolute levels were still low. Analysis of PA precursors using LC-MS indicated that, compared with untransformed B104 seeds, where these precursors are barely detectable, the levels of epicatechin, 4β-(S-cysteinyl)-catechin, and 4β-(S-cysteinyl)-epicatechin were increased by over 100-fold in SbTT2-OX and to a lesser extent in SbMYB5-OX (Fig. 5c, Supplementary Fig. 11). In contrast to the ST-GmANR1 and OSA-GmANR1 lines, where 4β-(S-cysteinyl)-epicatechin content is higher than 4β-(S-cysteinyl)-catechin, SbTT2-OX and

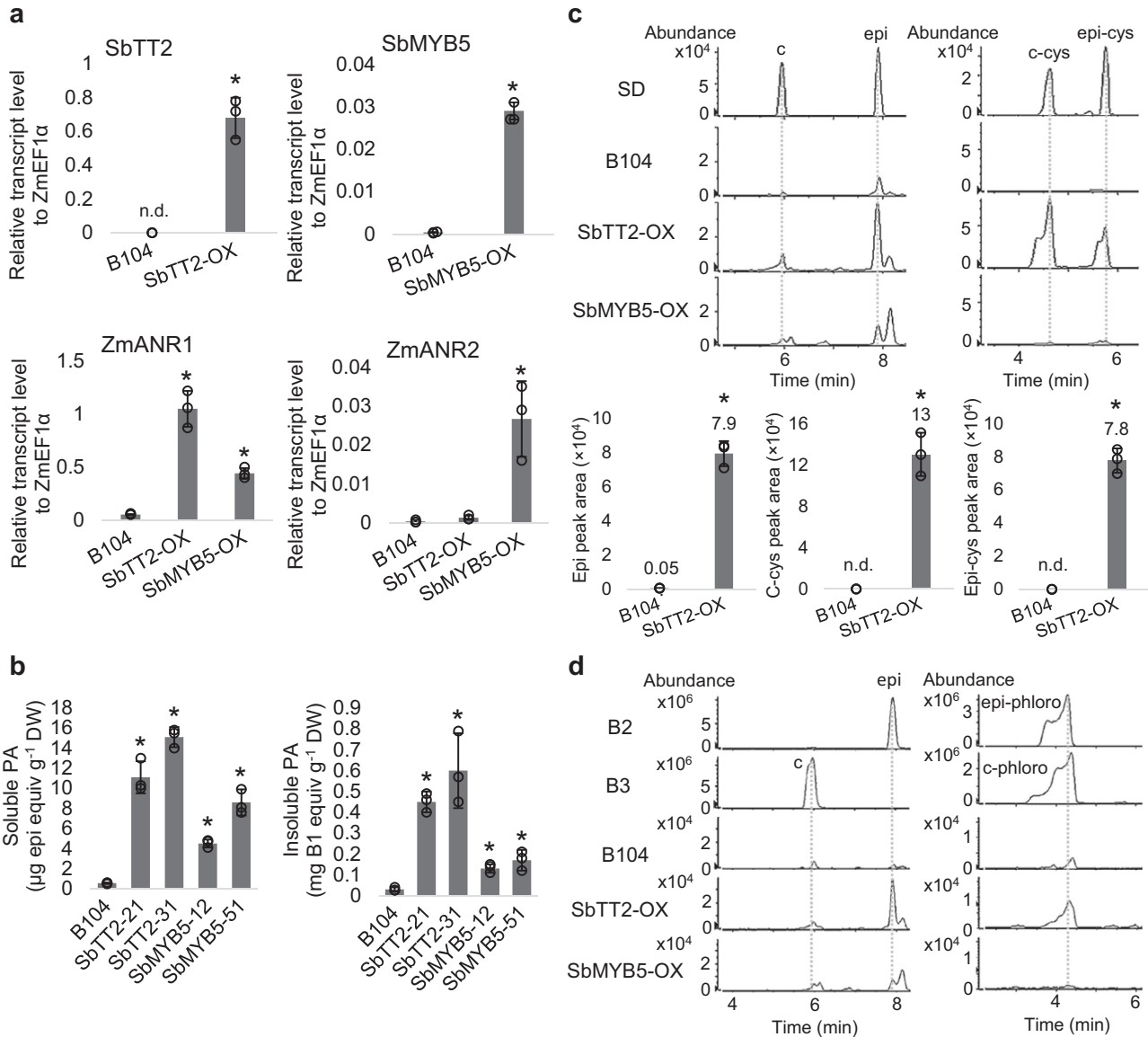

**Fig. 5 | PA and PA precursor content and composition in seeds of B104 maize expressing *SbTT2* or *SbMYB5*. a** Transcript levels of *SbTT2*, *SbMYB5*, *ZmANR1* and *ZmANR2* in developing seeds (14-days after pollination) of untransformed B104, SbTT2-OX and SbMYB5-OX analyzed by qRT-PCR. Data are presented as mean ± S.D. (*n* = 3, independent biological replicates). *ZmEF1α* was used as reference gene. Asterisks indicate significant difference relative to the untransformed control at *P* < 0.01 as determined by two-tailed Student's *t*-test. **b** Contents of soluble and insoluble PAs in seeds of B104, SbTT2-OX and SbMYB5-OX. Data are presented as mean ± S.D. (*n* = 3, independent biological replicates). Asterisks indicate significant difference relative to the untransformed control at *P* < 0.01 as determined by two-tailed Student's *t*-test. **c** Selected ion chromatograms of catechin and epicatechin (m/z = 289.0718 ± 10 ppm), as well as 4*β*-(S-cysteinyl)-catechin and 4*β*-(S-cysteinyl)- epicatechin (m/z = 408.0759 ± 10 ppm) in seeds of B104, SbTT2-OX (line SbTT2-31) and SbMYB5-OX (line SbMYB5-51). Peak areas of epicatechin (epi), 4*β*-(S-cysteinyl)- catechin (c-cys), and 4*β*-(S-cysteinyl)-epicatechin (epi-cys) in B104 and SbTT2-OX are shown in the histograms. Data are presented as mean ± S.D (*n* = 3, independent biological replicates). Asterisks denote significant difference relative to untrans- formed B104 at *P* < 0.01 determined by two-tailed Student's *t*-test. Compounds not detected are labeled as n.d. **d** Phloroglucinolysis of PAs extracted from seeds of B104, SbTT2-OX and SbMYB5-OX using procyanidin dimers B2 and B3 as refer- ences. Left, selected ion chromatograms of catechin and epicatechin monomers (m/z = 289.0718 ± 10 ppm); right, selected ion chromatograms for detection of catechin-phloroglucinol and epicatechin-phloroglucinol (m/z = 413.0876 ± 10 ppm). Source data for Figs. 5a–c are provided in the Source Data file.

untransformed maize (BM, ST, and OSA) contain about twice as much 4*β*-(S-cysteinyl)-catechin as 4*β*-(S-cysteinyl)-epicatechin (Figs. 3a, 3c, 5c), which is likely due to the endogenous ZmANR not being able to efficiently produce 2,3-*cis*-leucocyanidin[23]. Phloroglucinolysis assays showed that the amount of (epi)catechin-phloroglucinol produced was higher in SbTT2-OX lines than in untransformed plants (Fig. 5d), indicating increased accumulation of PA oligomers and polymers in SbTT2-OX seeds. It is likely that, due to the inefficiency of ZmANR, excess substrates are re-directed toward anthocyanin biosynthesis, resulting in hyperaccumulation of anthocyanins.

## Maize produces procyanidin dimers with unusual stereochemistry

In contrast to most ANRs, ZmANR1 converts cyanidin to (+)-epica- techin in vitro, which can subsequently form procyanidin dimers with 2,3-*cis*-leucocyanidin or 4*β*-(S-cysteinyl)-epicatechin[23]. Unlike (+)-epi- catechin and (-)-epicatechin monomers that cannot be separated on LC-MS, the iso-B2 dimer ((-)-epicatechin-(+)-epicatechin] and iso-B4 dimer [(+)-catechin-(+)-epicatechin] have different retention times than B2 [(-)-epicatechin-(-)-epicatechin] and B4 [(+)-catechin-(-)-epica- techin]. To test whether maize seeds contain different stereoisomers

of the common procyanidin dimers, standards for iso-B2 and iso-B4 dimers were chemically synthesized (Supplementary Fig. 12), and procyanidin dimers in polyphenol extracts from maize seeds were analyzed by LC-MS (see Methods). In ST maize seeds, we detected iso-B2 and iso-B4 as the major procyanidin dimers, both of which use (+)-epicatechin as starter unit and have not, to the best of our knowledge, been detected in any other plant species (Fig. 6a). These dimers could also be detected in seeds of BM and to a lesser extent AREQ, which contains less free epicatechin than BM and ST (Supplementary Fig. 13, Fig. 2). The level of iso-B4 tended to be higher than iso-B2 in these seeds, consistent with maize seeds preferentially containing more cysteinyl-catechin than cysteinyl-epicatechin (Fig. 2).

In seeds of ST-GmANR1 and OSA-GmANR1, in addition to iso-B2 and iso-B4 dimers, we also detected procyanidin B2 and B4 dimers (Fig. 6a, b), which use (-)-epicatechin as starter unit and are commonly found in soybean and other PA-rich species, supporting the idea that ANR determines the C2-C3 stereochemistry of procyanidin dimers. While the yellow-colored seeds of B104 do not contain detectable procyanidin dimers, overexpression of *SbTT2* in B104 induced production of iso-B2 and iso-B4 in seeds (Fig. 6c), similar to untransformed BM and ST varieties. Furthermore, procyanidin trimer C1 (epicatechin-epicatechin-epicatechin) was also detected in seeds of ST-GmANR1 using C1 trimers extracted from soybean and *M. truncatula* seeds as reference standards (Fig. 6d).

### Crosstalk between PA and other flavonoid pathways in maize

Although ectopic expression of *SbTT2* or *SbMYB5* induced production of PAs in maize, their content remained low compared to that in sorghum and other PA-rich plants, suggesting that maize may lack a mechanism to efficiently divert flux toward PA biosynthesis. Therefore, we speculated that upstream substrates may be routed to the synthesis of other flavonoid compounds.

Flavanol-anthocyanin conjugates (flavan-3-ol-anthocyanin) are dimeric flavonoids composed of a flavan-3-ol moiety and an anthocyanin moiety, with anthocyanin serving as the starter unit and flavan-3-ol as the extension unit. These compounds have been characterized in red wine, Guatemalan bean, blackcurrant and strawberry[39–44]. Over the past decade, several studies have also confirmed the existence of flavanol-anthocyanin dimers (i.e., catechin-cyanidin-3,5-O-diglucoside) in some purple maize seeds[21,45–47]. Due to the lack of commercial standards, we used catechin-cyanidin-3,5-O-diglucoside (Ca-C35G, m/z = 897.2127 ± 10 ppm) extracted from seeds of AREQ maize as a reference, as identified in a previous study[47]. In this way, we were able to detect catechin-cyanidin-3,5-O-diglucoside in purple (ST) but not in yellow (B104) or blue (BM) maize seeds (Supplementary Fig. 14a, b). Among the samples, AREQ had the highest levels of catechin-cyanidin-3,5-O-diglucoside and cysteinyl-catechin, the proposed extension unit for the conjugates (Fig. 2). In addition, overexpression of *SbTT2* or *SbMYB5* in maize with yellow seeds (B104) induced production of catechin-cyanidin-3,5-O-diglucoside in the seeds (Supplementary Fig. 14c). Interestingly, even though anthocyanins and PAs were present at higher levels in SbTT2-OX than in SbMYB5-OX, SbMYB5-OX produced more catechin-cyanidin-3,5-O-diglucoside than SbTT2-OX (Supplementary Fig. 14c), likely due to the increased flux of flavonoid substrates.

To further explore the relationship between anthocyanin and PA biosynthesis, we analyzed PAs in a classic maize mutant line *bz1*, where anthocyanin biosynthesis and deposition are blocked due to the deficiency of a UDP-glucose: flavonoid 3-O-glucosyltransferase (UFGT), a key enzyme in anthocyanidin glycosylation[48,49]. The *bz1* seeds had much higher levels of epicatechin but dramatically lower levels of cyanidin 3-O-glucoside than *BZ1* seeds (Fig. 7a), likely because ANR and UFGT compete for common substrates and the disruption of UFGT results in substrates being redirected to PA biosynthesis. In fact, the accumulation of anthocyanins was almost abolished in *bz1* seeds

(Fig. 7b). The relatively high levels of epicatechin in *bz1* seeds allowed us to determine its stereochemistry using chiral-HPLC, confirming the (+)-epicatechin stereoisomer (Fig. 7c). In contrast, (-)-epicatechin is the only epicatechin stereoisomer detected in soybean seeds (Supplementary Fig. 15), likely generated by the LAR-LDOX-ANR complex[23] and consistent with the result that only procyanidin B2 [(-)-epicatechin-(-)-epicatechin] was present in soybean seed coats.

Levels of procyanidin dimers, iso-B2 and particularly iso-B4, were higher in *bz1* seeds compared with purple *BZ1* seeds with the natural *UFGT* allele (Fig. 7d). The higher level of iso-B4 than iso-B2 in *bz1* seeds is consistent with *bz1* accumulating more cysteinyl-catechin than cysteinyl-epicatechin (Fig. 7a). Phloroglucinolysis confirmed the increased epicatechin levels, but the low relative level of released (epi) catechin-phloroglucinol in *bz1* relative to *BZ1* seeds (Supplementary Fig. 16) suggests that epicatechin extension units were not present in longer PA polymers.

## Discussion

Maize is known for having great diversity of flavonoids across cultivars, associated with seed coloration. However, although anthocyanin and phlobaphene biosynthesis pathways in maize have been well established, oligomeric PAs appear to be absent in this species. The biosynthetic pathways of anthocyanins, phlobaphenes, PAs, and other flavonoids partially overlap, sharing several common precursors and enzymes (Fig. 8)[1,8]. Here we show that, among different maize cultivars with an active anthocyanin biosynthetic pathway, those producing less anthocyanins tend to accumulate more epicatechin and other PA precursors. The increased PA precursor accumulation in *bz1* maize deficient in the enzyme converting cyanidins into anthocyanins suggest that PA biosynthesis may be closely related to the flux into anthocyanin biosynthesis. It is less clear whether the same is true with regard to phlobaphenes, the biosynthesis of which diverts prior to the anthocyanin pathway and is localized to pericarp tissues (Fig. 8). Despite the lack of an apparent homolog of TT2, a transcription factor specifically activating PA biosynthesis in PA-rich plant species[34], overexpression of *SbTT2* in maize induced the production of PAs. Surprisingly though, expression of *SbTT2* in a maize inbred with an inactive anthocyanin biosynthesis pathway also led to anthocyanin accumulation, resulting in purple-colored seeds in transgenic lines. Previous studies demonstrated that *C1* and *R* family transcription factors regulate anthocyanin biosynthesis in maize with dark-colored seeds[15,50]. The PA precursor and anthocyanin accumulation in seeds of SbTT2-OX transgenic lines resembled that of maize cultivars with functional transcription factors for anthocyanin biosynthesis, supporting the hypothesis that SbTT2 and maize anthocyanin-related transcription factors share conserved activities in regulating PA and anthocyanin biosynthesis in maize.

ANR is a key enzyme that produces both PA starter and extension units, and ANR product specificity determines PA composition and polymerization[23]. Our previous in vitro studies showed that unlike ANRs from PA-rich species such as soybean that can efficiently produce (-)-epicatechin via coupled reactions with LDOX (absent in maize) using (+)-catechin as substrate, ZmANR1 produces (+)-epicatechin at low efficiency in such in vitro LDOX-coupled reactions and favors the production of (+)-epicatechin over other flavan-3-ol stereoisomers when using cyanidin as substrate[23]. Our biochemical analyses of PAs in maize show that (+)-epicatechin is the epicatechin stereoisomer detected in *bz1* seeds, and procyanidin dimers with (+)-epicatechin as starter units (i.e., iso-B2 and iso-B4 dimers) are the predominant procyanidin dimers in seeds of SbTT2-OX, where endogenous *ZmANR1* expression was upregulated. Seeds of ST-GmANR1 and OSA-GmANR1, on the other hand, produced classical procyanidin B2 and B4 dimers, which have (-)-epicatechin as starter units and are commonly found in PA-rich plant species. In addition, expression of *GmANR1* in ST or OSA maize dramatically increased 4β-(S-cysteinyl)-epicatechin level in

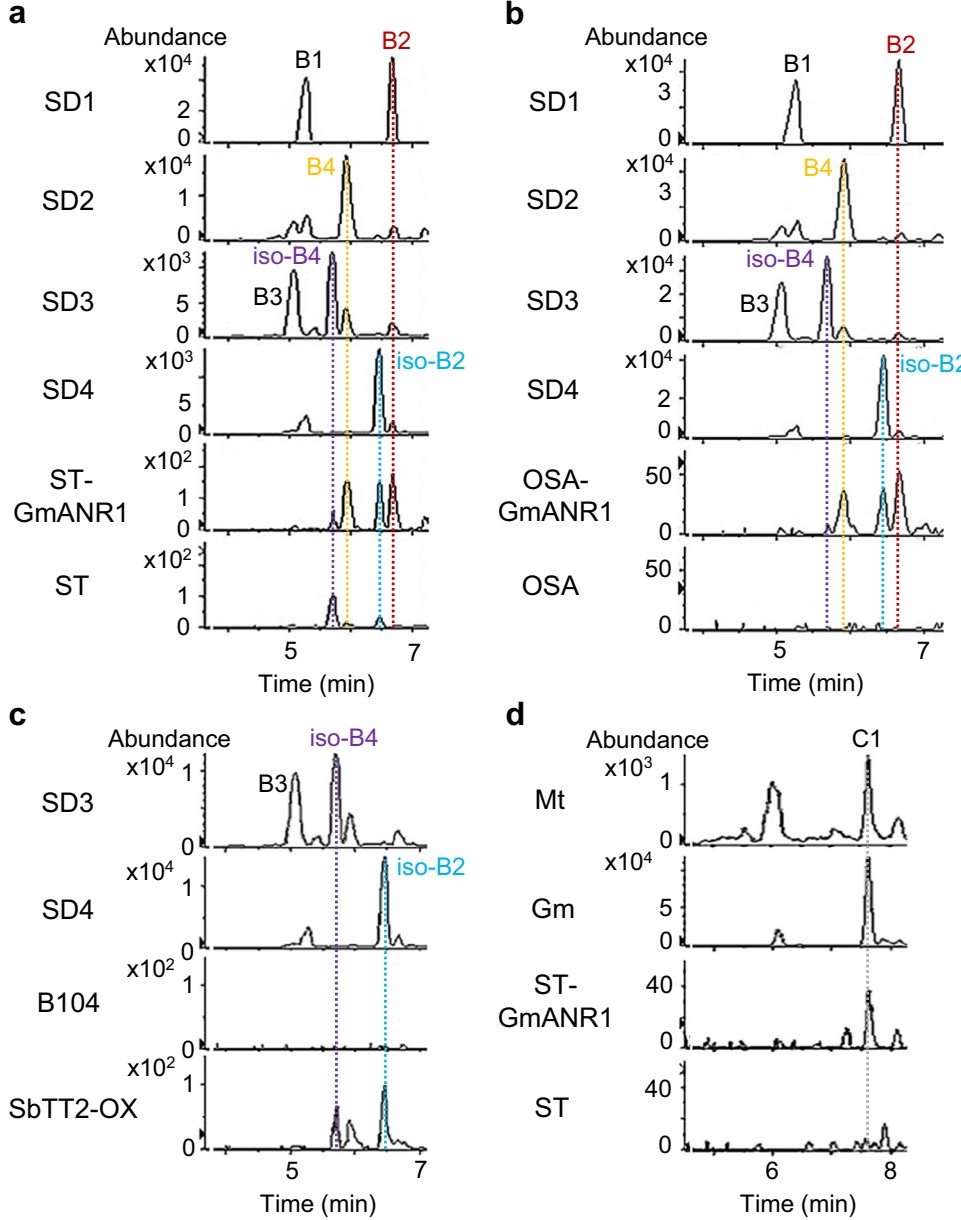

**Fig. 6 | Analysis of procyanidin dimers and trimers in maize seeds expressing *GmANR1* or *SbTT2*.** Selected ion chromatograms of procyanidin dimers in standards and PAs extracted from seeds of untransformed and transgenic maize ST (**a**), OSA (**b**), and B104 (**c**). **d** Selected ion chromatograms of procyanidin trimers in PAs extracted from seeds of Medicago (Mt), soybean (Gm), untransformed ST maize, and ST maize expressing *GmANR1*. SD, chemical standards; ST, Suntava maize; ST-GmANR1, transgenic maize seeds obtained from crosses between Hill-GmANR1 and ST; OSA, Osage maize; OSA-GmANR1, transgenic maize seeds obtained from crosses between Hill-GmANR1 and OSA; B1-B4, procyanidin dimers B1-B4 (m/z = 577.1360 ± 10 ppm); iso-B2, procyanidin B2 isomer with (+)-epicatechin as the starter unit; iso-B4, procyanidin B4 isomer with (+)-epicatechin as the starter unit; C1, procyanidin C1 trimer (m/z = 865.1981 ± 10 ppm).

seeds, consistent with our previous finding that GmANR1 is directly involved in the formation of 4β-(S-cysteinyl)-epicatechin extension units[23]. Interestingly, while ZmANR1 only weakly catalyzes in vitro conversion of flaven-3,4-diol to 2,3-*cis*-leucocyandin and 4β-(S-cysteinyl)-epicatechin, trace amounts of 4β-(S-cysteinyl)-epicatechin were detected in seeds of some maize varieties and expression of *SbTT2* greatly enhanced its accumulation, suggesting that there may be other factors or pathways within maize cells contributing to the production of 4β-(S-cysteinyl)-epicatechin and that this mechanism can be activated by SbTT2.

Although expression of *GmANR1* in colored maize increased the levels of epicatechin and 4β-(S-cysteinyl)-epicatechin starter and extension units, the level of PA polymers remained low, suggesting an inefficient PA polymerization mechanism. Given that the assembly of

PA polymers is likely nonenzymatic[29], maize may lack an appropriate cellular mechanism to stabilize and protect reactive PA intermediates during delivery to appropriate cell compartments for polymerization. This might explain the presence of unusual (epi)catechin-anthocyanin conjugates in some maize varieties. The procyanidin dimers iso-B2 and iso-B4 were also detected in some maize varieties and in transgenic maize overexpressing *SbTT2*, suggesting that the (+)-epicatechin produced by ZmANR1 can be used as starter unit to generate procyanidin dimers. It will now be important to investigate whether the unusual (+)-epicatechin and isomeric procyanidin dimers negatively influence PA assembly.

The scheme in Fig. 8 summarizes what is currently known about PA biosynthesis in maize. In the apparent absence of an efficient polymerization system, the operation of an ANR that generates the 2,3-

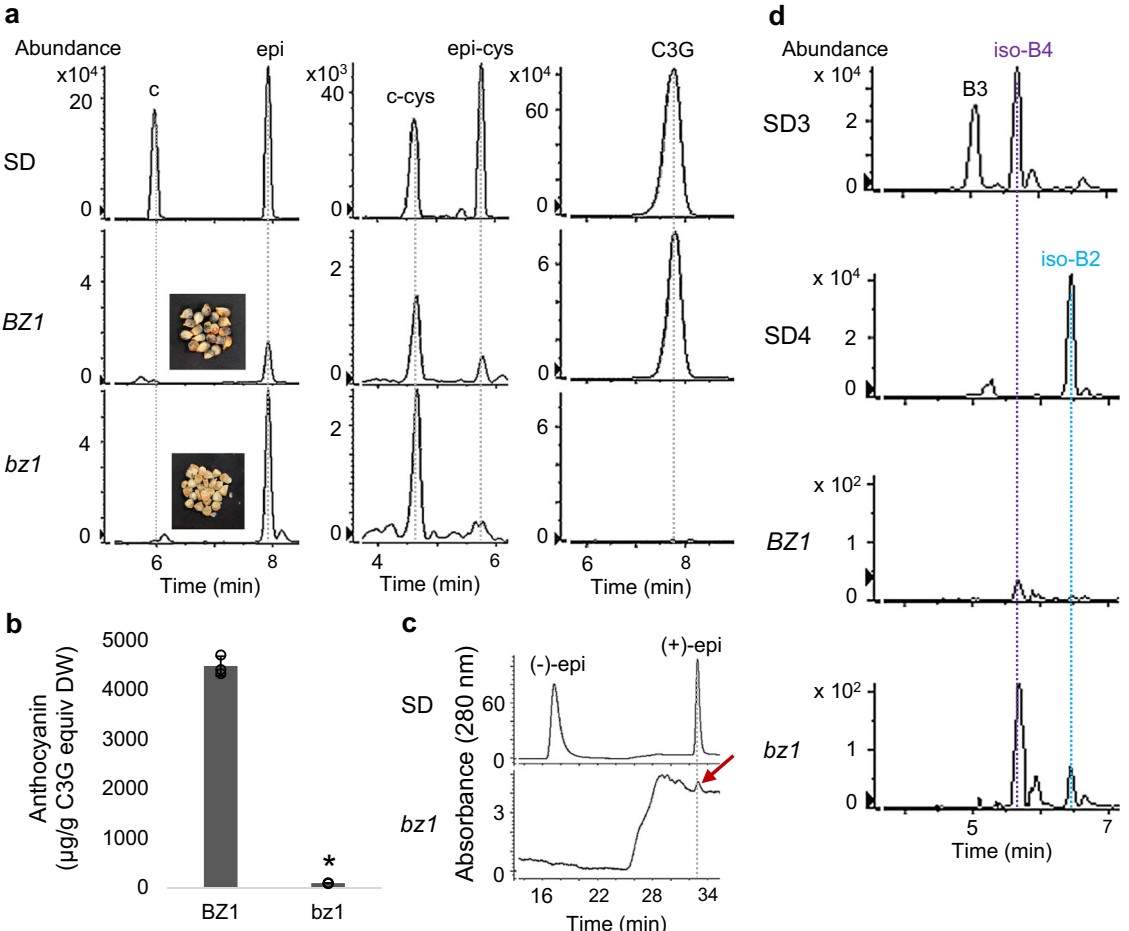

**Fig. 7 | Analysis of anthocyanins and PA precursors in seeds of *BZ1* and *bz1* mutant maize. a** Selected ion chromatograms of catechin and epicatechin (left, m/z = 289.0718 ± 10 ppm), 4β-(S-cysteinyl)-catechin and 4β-(S-cysteinyl)-epicatechin (middle, m/z = 408.0759 ± 10 ppm), as well as cyanidin 3-*O*-glucoside (right, m/z = 447.0940 ± 10 ppm) in seeds of *BZ1* and *bz1* maize. **b** Anthocyanin contents in *BZ1* and *bz1* seeds. Data are presented as mean ± S.D. (*n* = 3, independent biological replicates). Asterisks indicate significant difference between *BZ1* and *bz1* samples at *P* < 0.01 as determined by two-tailed Student's *t*-test. **c** Chiral-HPLC analysis of the stereochemistry of epicatechin monomers in *bz1* seeds. Red arrow indicates the (+)-epicatechin in PAs extracted from *bz1* seeds. **d** Selected ion chromatograms of procyanidin dimers (m/z = 577.1360 ± 10 ppm) in *BZ1* and *bz1* maize seeds. SD, chemical standard; (-)-epi and (+)-epi, (-)-epicatechin and (+)-epicatechin; c-cys and epi-cys, 4β-(S-cysteinyl)-catechin and 4β-(S-cysteinyl)-epicatechin; C3G, cyanidin 3-*O*-glucoside; B3, procyanidin B3 dimer; iso-B2, procyanidin B2 isomer with (+)-epicatechin as the starter unit; iso-B4, procyanidin B4 isomer with (+)-epicatechin as the starter unit. Source data for Fig. 7b are provided in the Source Data file.

trans-flavan-3,4-diol, and a route to catechin-cysteine via 2,3-trans-leucocyanidin, the major products are isomers of procyanidins B2 and B4, along with catechin-anthocyanin conjugates. The recent publication of a set of candidate PA biosynthesis genes expressed in the seed coat of wheat[51], which naturally accumulates (+)-catechin type PAs, will facilitate comparative approaches to deciphering the key controlling steps in PA biosynthesis in cereals. Engineering PAs in maize represents a promising strategy for producing value-added agricultural products and for improving the health benefits of maize for humans and livestock, and understanding the unconventional endogenous pathway in maize sets the stage for this purpose.

## Methods
### Chemicals and seed stocks
Chemical standards of (+)-catechin, (-)-epicatechin, cyanidin chloride and cyanidin 3-*O*-glucocide were purchased from Sigma-Aldrich. The standard of (+)-epicatechin was purchased from Nacalai USA. Standards of 4β-(S-cysteinyl)-catechin and 4β-(S-cysteinyl)-epicatechin were synthesized using procyanidin dimers B3 or B2 and L-cysteine as described previously[29]. FBLL (PI 546481), Black Mexican (PI162573) and Arequipa (PI 571427) seeds were ordered from USDA. Suntava and Osage seeds were ordered from Burpee Gardens and

Baker Creek Heirloom Seeds, respectively. Maize seeds rich in phlobaphene (*P1-rr*) in the A619 background were provided by Drs. Nan Jiang and Erich Grotewold at Michigan State University. Tobacco (*Nicotiana tabacum*, cv *Xanthi*) seeds constitutively expressing *AtPAP1* were obtained from Dr. De-Yu Xie at North Carolina State University. *BZ1* and *bz1* maize seeds both in the W23 (*C1, R-r*) genetic background were provided by Dr. Virginia Walbot at Stanford University.

### Plant growth conditions and generation of transgenic plants
Plants of maize, tobacco and soybean were grown in greenhouses with temperature and light conditions set at 25 °C/22 °C day/night, and 14 h/10 h of light/dark. The coding sequence of GmANR1 was cloned from soybean (cv *Clark*) as previously described[24]. The coding sequences of SbTT2 and SbMYB5 were synthesized at Gene Universal. Sequences encoding GmANR1, SbTT2 and SbMYB5 were individually cloned in pMCG1005 vector between *Asc*I and *Xma*I restriction enzyme sites and driven by the maize Ubiquitin promoter. *Agrobacterium*-mediated maize transformation was carried out by the Plant Transformation Facility at Iowa State University[52]. *GmANR1* was transformed into HiII maize, and F₁ seeds were obtained by self-pollination of transgenic plants. *SbTT2* and *SbMYB5* were transformed into B104

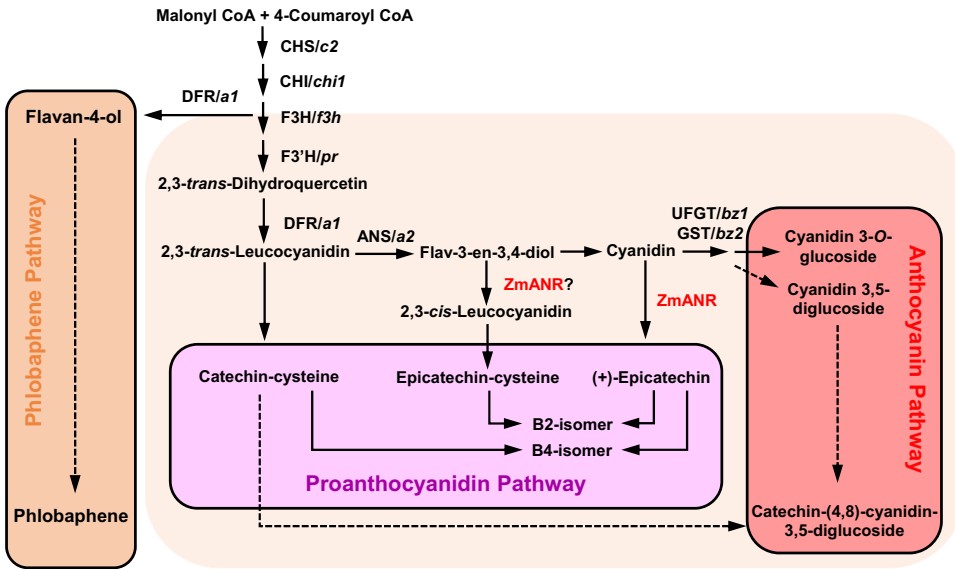

**Fig. 8 | Schematic diagram of PA, anthocyanin and phlobaphene biosynthesis pathways in maize seeds.** CHS, chalcone synthase, *c2;* CHI, chalcone isomerase, *chi1*; F3H, flavanone 3-hydroxylase, *f3h*; F3'H, flavanone 3´-hydroxylase, *pr1*; DFR, dihydroflavanol 4-reductase, *a1*; ANS, anthocyanidin synthase, *a2*; ANR, anthocyanidin reductase; UFGT, UDP-glucose: flavonoid glucosyltransferase, *BZ1*; GST, glutathione S-transferase, *BZ2*. Anthocyanins accumulate in aleurone cells, and phlobaphenes in the pericarp.

maize, and $F_1$ seeds were obtained by pollinating transgenic plants with pollen collected from untransformed B104 plants.

Transgenic soybean (cv *Clark*) hairy roots were generated using an *Agrobacterium*-mediated transformation method[53]. Briefly, coding sequences of SbTT2, SbMYB5 and GUS were cloned into pB7WG2D gateway vector driven by p35S promoter, and the recombinant plasmids were subsequently transformed into *Agrobacterium rhizogenes* strain K599. Cotyledons from soybean (cv. *Clark*) were used for transformation and transgenic roots were maintained in B5 medium and sub-cultured every 3 weeks. For tobacco transformation, coding sequences of AtANR, GmANR1 and ZmANR1 were cloned into pB7WG2D gateway vector, and the recombinant plasmids were transformed into *Agrobacterium tumefaciens* strain LB4404. Details of subsequent transformation were as described previously[54].

### Enzymatic assays

*GmANR1* and *ZmANR1* were individually cloned in pDEST17 Gateway destination vector and transformed in Rosetta™ (DE3) cells (Millipore Sigma). Protein extraction and purification were based on protocols described previously[55]. Enzymatic assays were carried out in a total volume of 100 μL containing 0.1 mM cyanidin, 1 mM NADPH, 50 mM MES buffer pH 5.7 and protein (50 μg) at 42 °C for 1 h and stopped by adding 200 μL ethyl acetate.

### Transactivation assays and protein subcellular localization

Transactivation assays using protoplasts isolated from Arabidopsis leaves were performed following the protocol described previously[56], and firefly luciferase activity was quantified as previously described[37]. For subcellular localization, coding sequences of GmANR1 and ZmANR1 were inserted into pMDC43 gateway vector. The recombinant plasmids with GFP-tagged GmANR1 and ZmANR1 were then introduced into Arabidopsis protoplasts to visualize the subcellular localization of recombinant proteins. Confocal images of protoplasts expressing the GFP-tagged proteins were acquired by a Zeiss LSM710 confocal laser-scanning microscope. GFP was excited by a 488 nm laser and emission signals were collected at 500–540 nm.

### RNA extraction and qRT-PCR analysis

RNA samples were extracted from maize leaves, seeds and soybean hairy roots using PureLink Plant RNA Reagent (Invitrogen), and then treated with DNA-*free* DNA removal kit (Invitrogen) according to the manufacturers' manuals. Then, cDNA was synthesized using iScript Select cDNA synthesis kit (Bio-Rad). Gene expression (transcript) levels were analyzed by qRT-PCR using PowerUP SYBR Green Master Mix (Applied Biosystems) and a QuantStudio 6 Flex Real-Time PCR system. Sequences of primers used in this study are listed in Supplemental Table 1.

### Extraction and quantification of anthocyanins and PAs

Anthocyanins were extracted from maize seeds by adding 0.5 mL of 0.1% HCl in methanol to about 10 mg of ground seed powders, sonicating for 1 h, and shaking overnight in a dark cold-room. Samples were then centrifuged at 13,000 rpm for 5 min, and supernatants were transferred to new tubes. Anthocyanins were further fractionated by adding equal volumes of water and 0.5 mL of chloroform and centrifuging at 13,000 rpm for 5 min. Total anthocyanins in the upper aqueous phase were quantified based on the absorbance at 520 nm using a standard curve generated with cyanidin 3-*O*-glucoside. Hydrolysis of anthocyanins was performed by heating under acidic conditions (2 N HCl) at 100 °C for 1 h.

Soluble PAs were extracted by adding 1 mL of extraction buffer (70% acetone, 0.5% acetic acid) to 100–200 mg of ground whole maize seeds and sonicating for 1 h at room temperature. Samples were centrifuged at 13,000 rpm for 5 min, and supernatants transferred to new tubes. PAs in the supernatants were further fractionated by washing with equal volumes of chloroform three times and washing with hexane once. The cleaned PA samples were dried under vacuum and dissolved in 50% methanol. PAs were quantified by reaction with DMACA and measuring the absorbance at 640 nm. Insoluble PAs were quantified using the butanol-HCl method as described previously[29].

The distribution of insoluble PAs in soybean hairy roots was determined by staining cross sections with DMACA staining buffer (0.2% DMACA in methanol/$H_2O$ containing 3 N HCl) overnight and washing repeatedly with 70% ethanol prior to imaging. Phloroglucinolysis was performed by adding 50 μL of fresh phloroglucinol

solution (5% phloroglucinol, 1% ascorbic acid, 1 N HCl dissolved in methanol) to dried PA extract and incubating at 37 °C for 20 min. The reaction was stopped by adding 50 μL of sodium acetate (0.2 M) and immediately subjecting to LC-MS analysis.

## Analysis of anthocyanins and PAs by HPLC and LC-MS

Anthocyanins were analyzed using an Agilent 1290 Infinity II HPLC system equipped with an Eclipse Plus C18 column (4.6 × 250 mm, 5 μm). The samples were separated at a rate of 1 mL/min using solvent A (0.1% formic acid) and solvent B (acetonitrile) with the following protocol: 0 to 5 min, 5% B; 5 to 10 min, 5–10% B; 10 to 25 min, 10–17% B; 25 to 40 min, 17–50% B; 40 to 50 min, 50–90% B; 50 to 60 min, 90–100% B. Data were collected at 520 nm.

Chiral HPLC analysis of epicatechin was performed using an Agilent HP1100 system equipped with a chiral column (Chiral Technologies #80325) with solvent A (0.5% acetic acid in hexane) and solvent B (0.5% acetic acid in ethanol) at 1 mL/min. The protocol used was as follows: 0 to 20 min, 20% B; 20 to 23 min, 20–50% B; 23–38 min, 50% B; 38–40 min, 50–20% B. Signals were recorded at 280 nm.

Soluble PAs were analyzed with an Exion ultra high-performance liquid chromatography system coupled with a high resolution Triple-TOF6600+ mass spectrometer from AB Sciex as described previously[23,24]. Specifically, compounds were separated using a C18 Acquity UPLC HSS T3 (100 × 2.1 mm, 1.8 μm) column from Waters. The column compartment and the autosampler temperatures were kept at 42 °C and 15 °C, respectively. The analytes were eluted using a gradient of 0.1% formic acid in water (Solvent A) and 0.1% formic acid in methanol (Solvent B) under a flow rate of 0.4 mL/min. The following gradient was applied: 0–1.0 min, 5.0 % B; 1.0–2.0 min, 5.0–10.0% B; 2.0–7.0 min, 10.0–28.2% B; 7.0–11.0 min, 28.2–70.0% B; 11.0–11.1 min, 70.0–95.0% B; 11.1–13.0, 95% B; 13.0–13.1 min, 95.0–5.0% B; 13.1–15.0 min, 5.0% B. In short, the mass spectrometer was set to scan metabolites from m/z 250–1000 amu in negative mode with an ion spray voltage of 4000 V and the accumulation time was 100 msec. MS/MS spectra were acquired over m/z 30–1000 amu with an accumulation time of 25 msec and parameters such as declustering potential, collision energy and collision energy spread were set to 50 V, 25 V and 10 V, respectively. The temperature of the source was 500 °C. The analysis of the data was performed using Sciex OS software as described previously[23].

## Extraction of polyphenols from maize seeds

To analyze procyanidin dimers and trimers in maize seeds, a modified polyphenol extraction and purification method was used. Briefly, maize seeds were ground into fine powders using liquid nitrogen and 1 mL of 80% methanol was added to 100–200 mg homogenized samples. After sonication for 1 h, samples were centrifuged for 5 min at 13,000 rpm. Supernatants were transferred to new tubes and dried under vacuum. Next, 100 μL of water was added to each sample, and 200 μL of ethyl acetate was then added, followed by vortexing and centrifugation at 13,000 rpm for 5 min. The upper ethyl acetate phases were transferred to new tubes and dried under vacuum. Finally, the samples were dissolved in 20 μL of 50% methanol and stored at −20 °C until use.

## Phylogenetic analysis

Multiple protein sequence alignments were performed using the ClustalW program, and the phylogenetic tree was generated by the MEGA program following the Neighbor-Joining method with 2000 bootstrap replicates[57].

## Statistical analysis

Significant differences (P < 0.05, 0.01) were determined by Student's t-test. All statistical analyses were performed using three to five biological replicates, and n values in different experiments are listed in figure legends. Exact p-values are listed in the Source Data file wherever applicable.

## Reporting summary

Further information on research design is available in the Nature Portfolio Reporting Summary linked to this article.

## Data availability

Materials generated and analyzed in this study are available from the corresponding author upon request. A reporting summary for this paper is available as a Supplementary information file. The LC-MS data used in this study are available in the MetaboLights database[58] under accession code MTBLS7920. Source data are provided with this paper.

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

## Acknowledgements
This work was funded by a grant to RAD from Grasslanz Technology Limited, Palmerston North, New Zealand. We thank Dr. Chenggang Liu for advice in synthesizing the chemical standards, and the BioAnalytical Facility at the University of North Texas for metabolite analysis.

## Author contributions
N.L. and R.A.D. designed the experiments. N.L. and Y.L. performed the experiments. N.L., J.H.J. and R.A.D. analyzed data. N.L. and R.A.D. wrote the manuscript.

## Competing interests
The authors declare no competing interests.
