## [Peer Review File · Nature Communications]

REVIEWERS' COMMENTS

Reviewer #1 (Remarks to the Author):

Lu et al reported an investigation of proanthocyanidin (PA) biosynthesis in maize, one of main food crops. Because PAs and their monomeric precursors, flavan-3-ols, are important nutraceuticals, understanding their biosynthesis and structures is necessary to create value-added maize for human and animal healths.

The authors mined the genomes of maize and other monocotyledon plants and found two ANR homologs with different profiles of them in tissues and C1- and MYB6-type TFs. Profiling different maize varieties revealed a different patterns of cysteinyl-(+)-catechin and (-)-epicatechin two types of extension unit precursors from other species. To characterize this phenomenon, they introduced GmANR1, SbTT2, and SbMYB5 to different maize varieties. Detailed analyses of PAs and their monomeric starter and extension units revealed that the maize ANR converted cyanidin to (+)-epicatechin and maize plants produced iso-B2 and -B4 structures with (+)-epicatechin as the starter unit.

Although the biosynthesis of PAs has gained well characterized by the authors' lab and others, this finding is new and important to fully understand the biosynthesis of PAs in plants. Therefore, data are valuable.

The English writing is excellent and easy to follow. Methods are appropriate and solid. Data are solid.

Minors concerns:

- 1, in the authors' lab and in 2013, Pang et al reported " Functional Characterization... . Plant Physiology. In Figure 3 I and J showed green ANR1 and 2 could convert cyanidin to (+)-catechin, (-)-catechin, (-)-epicatechin and (+)-epicatechin. Did you see procyanidins with (+)-epicatechin as starter unit in green tea?
- 2, in your methods, you used an acidic condition to extract flavan-3-ols and PAs. As well known, acids can epimerize flavan-3-ols at C2 and C3. Is there a possibility that (+)-epicatechin resulted from acid-based epimerization?
- 3, lines 217-227, the cis-elements bound by SbTT2 and SbMYB5 should be provided here. How long was the promoter cloned? What other cis-elements does it have?
- 4, lines 260-262, Figure 5 c and supplementary Fig. 11, the folders are unclear. labeling the area values on the top of bars helps readers to follow up the descriptions.
- 5, lines 262-264 describe OSA, ST, and BM, however, Fig. 5c does not have OSA and ST at all
- 6, line 278, seed should be a plural.
- 7, line 345, "biosynthesis" should be "biosynthetic" .
- 8, line 466, your anthocyanins are not purified. I believe that this method was the same as described by Xie et al 2006, Plant Journal from your lab. The same problem occurs on line 473 about PA.
- 9, lines 477-478, this overnight staining could only show cell wall-bound PAs but could not show you extractable PAs and flavan-3-ols.
- 10 line 516, "purify", this approach does not purify any compounds.
- 11, is this observation an orphan event or a consequence of evolution or domestication of maize?

Reviewer #2 (Remarks to the Author):

The manuscript by Lu et al entitled 'An unconventional proanthocyanidin pathway in maize' provides an advance in our understanding of an unconventional form of one of the most well-characterized natural product biosynthetic pathways in plants. The most highly consumed agricultural crop in the

world, namely *Zea mays*, was found to biosynthesize unconventional proanthocyanidins, which are polymeric natural products that have health benefits for humans and ruminant animals. The unconventional mechanism was traced to the ZmANR enzyme that preferentially catalyzes the formation of (+)-epicatechin, a monomeric subunit of proanthocyanidins that has uncommon stereochemistry. The unusual catalytic activity of the enzyme may also contribute to reduced polymerization of proanthocyanidins. Maize typically only synthesizes relatively low amounts of proanthocyanidins compared to other plant species and its genome was found to be missing one of the key transcription factors that regulates proanthocyanidin biosynthesis. Interestingly, ectopic expression of that gene from another species in maize dramatically increased proanthocyanidin biosynthesis, indicating that the regulatory network remained intact. It would have been interesting to learn whether the current form of ZmANR and missing TT2 transcription factor was found in wild relatives of maize and whether these unusual findings were a consequence of domestication. Yet, overall, this manuscript is expected to be of interest to the broad readership of *Nature Communications*.

Minor comments:

- What is the difference between varieties Suntava (ST) and Black Mexican (BM)? The two varieties are introduced to show different expression levels of ZmANR1 and ZmANR2, but what is the significance of these two varieties. Do they differ in proanthocyanidin content?
- Fig. S1, the blue arrows for LAR and LDOX may be misplaced. Both enzymes use leucocyanidin as a substrate and catalyze distinct branchpoints, LAR produces catechin and LDOX cyanidin.
- ZmANR1 has lower expression than AtANR and GmANR1 in tobacco. It should be mentioned why the low expression is not the reason for minimal formation of epicatechin, epi-cys, and procyanidin B2.

Response to reviewers

Reviewer #1 (Remarks to the Author)

Lu et al reported an investigation of proanthocyanidin (PA) biosynthesis in maize, one of main food crops. Because PAs and their monomeric precursors, flavan-3-ols, are important nutraceuticals, understanding their biosynthesis and structures is necessary to create value-added maize for human and animal health.

The authors mined the genomes of maize and other monocotyledon plants and found two ANR homologs with different profiles of them in tissues and C1- and MYB6-type TFs. Profiling different maize varieties revealed a different pattern of cysteinyl-(+)-catechin and (-)-epicatechin two types of extension unit precursors from other species. To characterize this phenomenon, they introduced GmANR1, SbTT2, and SbMYB5 to different maize varieties. Detailed analyses of PAs and their monomeric starter and extension units revealed that the maize ANR converted cyanidin to (+)-epicatechin and maize plants produced iso-B2 and -B4 structures with (+)-epicatechin as the starter unit.

Although the biosynthesis of PAs has been well characterized by the authors' lab and others, this finding is new and important to fully understand the biosynthesis of PAs in plants. Therefore, data are valuable.

The English writing is excellent and easy to follow. Methods are appropriate and solid. Data are solid.

Minor concerns:

1, in the authors' lab and in 2013, Pang et al reported "Functional Characterization... Plant Physiology. In Figure 3 I and J showed green ANR1 and 2 could convert cyanidin to (+)-catechin, (-)-catechin, (-)-epicatechin and (+)-epicatechin. Did you see procyanidins with (+)-epicatechin as starter unit in green tea?

Response: That is definitely an interesting question but since we did not use tea as material in our project, we don't have the answer to that question. The *in vitro* activity of ANR with cyanidin does not necessarily reflect its *in vivo* activity. For example, Arabidopsis ANR also produces more than one product *in vitro* from cyanidin, but the PAs made *in vivo* are believed to consist only of (-)-epicatechin. Whether a plant contains (+)-epicatechin or (-)-epicatechin as PA starter unit also depends on the existence and expression levels of LAR and/or LDOX.

2, in your methods, you used an acidic condition to extract flavan-3-ols and PAs. As well known, acids can epimerize flavan-3-ols at C2 and C3. Is there a possibility that (+)-epicatechin resulted from acid-based epimerization?

Response: We don't think the extraction method affects the stereochemistry of PA precursors. By using the same method, the flavan-3-ol extracted from soybean seed coat is (-)-epicatechin, as

shown in Fig S15. This is consistent with the fact that soybean contain procyanidin B2 as major PA dimer.

3, lines 217-227, the cis-elements bound by SbTT2 and SbMYB5 should be provided here. How long was the promoter cloned? What other cis-elements does it have?

Response: The promoter cloned was about 1 kb. The binding of TT2- or MYB5-type MYB genes to ANR has been well documented (Gesell et al., 2014). Besides, we have shown that the cloned soybean ANR1 promoter can be activated by TT2-type MYBs in our previous publication (Lu et al., 2021). If we had not seen transactivation, it would be necessary to determine whether the problem was with the TF or the target promoter.

4, lines 260-262, Figure 5 c and supplementary Fig. 11, the folders are unclear. labeling the area values on the top of bars helps readers to follow up the descriptions.

Response: The values have been added to Fig 5c and Fig S11 as suggested. We have added the values for peak intensity on the traces to indicate how we arrived at the values for fold-change. Where no peak is detected in controls, the fold-change is obviously extremely large but not quantifiable- our description of > 100-fold is correct, however.

5, lines 262-264 describe OSA, ST, and BM, however, Fig. 5c does not have OSA and ST at all

Response: Thanks for pointing this out, OSA and ST numbers were in Fig 3a and 3c. We have added Fig 3a and 3c to this sentence.

6, line 278, seed should be a plural.

Response: Thanks for pointing this out. This has been corrected.

7, line 345, "biosynthesis" should be "biosynthetic".

Response: Thanks for pointing this out. It has been corrected.

8, line 466, your anthocyanins are not purified. I believe that this method was the same as described by Xie et al 2006, Plant Journal from your lab. The same problem occurs on line 473 about PA.

Response: We have changed “purified” to “fractionated” for both occasions.

9, lines 477-478, this overnight staining could only show cell wall-bound PAs but could not show you extractable PAs and flavan-3-ols.

Response: Correct. We now qualify with “the distribution of insoluble PAs”. The DMACA staining method was used to show the presence of PA in soybean hairy roots. In addition to the *in situ* DMACA staining, PAs in soybean roots were extracted and analyzed by spectrophotometry and LC-MS.

10 line 516, "purify", this approach does not purify any compounds.

Response: Correct. We have removed the phrase “to purify polyphenols”.

11, is this observation an orphan event or a consequence of evolution or domestication of maize?

Response: This is an interesting question but really difficult to answer. Based on our observation so far, PA precursors are more likely to accumulate in anthocyanin-rich maize seeds. More experiments are needed to draw any conclusion, but it is more likely evolutionary events than the other scenarios.

Reviewer #2 (Remarks to the Author)

The manuscript by Lu et al entitled ‘An unconventional proanthocyanidin pathway in maize’ provides an advance in our understanding of an unconventional form of one of the most well-characterized natural product biosynthetic pathways in plants. The most highly consumed agricultural crop in the world, namely *Zea mays*, was found to biosynthesize unconventional proanthocyanidins, which are polymeric natural products that have health benefits for humans and ruminant animals. The unconventional mechanism was traced to the ZmANR enzyme that preferentially catalyzes the formation of (+)-epicatechin, a monomeric subunit of proanthocyanidins that has uncommon stereochemistry. The unusual catalytic activity of the enzyme may also contribute to reduced polymerization of proanthocyanidins. Maize typically only synthesizes relatively low amounts of proanthocyanidins compared to other plant species and its genome was found to be missing one the key transcription factors that regulates proanthocyanidin biosynthesis. Interestingly, ectopic expression of that gene from another species in maize dramatically increased proanthocyanidin biosynthesis, indicating that the regulatory network remained intact. It would have been interesting to learn whether the current form of ZmANR and missing TT2 transcription factor was found in wild relatives of maize and whether these unusual findings were a consequence of domestication. Yet, overall, this manuscript is expected to be of interest to the broad readership of Nature Communications.

Minor comments:

- What is the difference between varieties Suntava (ST) and Black Mexican (BM)? The two varieties are introduced to show different expression levels of ZmANR1 and ZmANR2, but what is the significance of these two varieties. Do they differ in proanthocyanidin content?

Response: Thanks for the question. ST and BM represent different maize seeds that accumulate anthocyanins but show different colors, i.e. ST has purple seeds while BM has dark blue/black colored seeds. They both accumulate epicatechin and cysteinyl-catechin, though at different levels. According to Fig 2, BM has higher level of epicatechin but lower level of C3G. We now mention the seed colors.

-Fig. S1, the blue arrows for LAR and LDOX may be misplaced. Both enzymes use

leucocyanidin as a substrate and catalyze distinct branchpoints, LAR produces catechin and LDOX cyanidin.

Response: Sorry for the confusion. We were trying make the Figure as simple as possible, but in doing so have made it less so. In our 2000 paper in Nature plants (Jun et al referenced in the present paper), we showed that the pathway to epicatechin starter (but not extension) units in Medicago involves the formation of catechin from leucocyanidin catalyzed by LAR, followed by the conversion of catechin to cyanidin catalyzed by LDOX. This route does not occur in Arabidopsis. We now explain the two routes to epicatechin in Fig S1a in the Figure legend.

ZmANR1 has lower expression than AtANR and GmANR1 in tobacco. It should be mentioned why the low expression is not the reason for minimal formation of epicatechin, epi-cys, and procyanidin B2.

Response: Thanks for the question. The average expression levels of GmANR1 and AtANR are only 2-3 fold higher than ZmANR1, but the PA precursors in ZmANR1 transformed tobacco is barely detectable compared to the large increase of PA precursors in tobacco plants expressing GmANR1 or AtANR (Fig 1d). Besides, this result is consistent with our earlier paper where GmANR1 and AtANR can complement the *ban* mutant of Arabidopsis, while ZmANR1 only increased epicatechin monomer level, but not procyanidin B2 or PA polymer levels (Jun et al., 2021).

References:

Gesell, A., Yoshida, K., Tran, L.T., Constabel, C.P. Characterization of an apple TT2-type R2R3 MYB transcription factor functionally similar to the poplar proanthocyanidin regulator PtMYB134. *Planta* **240**, 497-511 (2014).

Jun, J.H., Xiao, X., Rao, X., Dixon, R.A. Proanthocyanidin subunit composition determined by functionally diverged dioxygenases. *Nat. Plants* **4**, 1034-1043 (2018).

Jun, J.H., Lu, N., Docampo-Palacios, M., Wang, X., Dixon, R.A. Dual activity of anthocyanidin reductase supports the dominant plant proanthocyanidin extension unit pathway. *Sci. Adv.* **7**, eabg4682 (2021).

Lu, N., Rao, X., Li, Y., Jun, J.H., Dixon, R.A. Dissecting the transcriptional regulation of proanthocyanidin and anthocyanin biosynthesis in soybean (*Glycine max*). *Plant Biotech. J.* **19**, 1429-1442 (2021).